# Molecular Characterization and Phylogenetic Analysis of *Spirometra* Tapeworms from Snakes in Hunan Province

**DOI:** 10.3390/vetsci9020062

**Published:** 2022-02-01

**Authors:** Shu-Yu Chen, Teng-Fang Gong, Jun-Lin He, Fen Li, Wen-Chao Li, Li-Xing Xie, Xin-Rui Xie, Yi-Song Liu, Ying-Fang Zhou, Wei Liu

**Affiliations:** 1Research Center for Parasites & Vectors, College of Veterinary Medicine, Hunan Agricultural University, Changsha 410128, China; ShuyuChen2021@stu.hunau.edu.cn (S.-Y.C.); GongTF@stu.hunau.edu.cn (T.-F.G.); hejunlin607@163.com (J.-L.H.); loislf@163.com (F.L.); Leo0725@stu.hunau.edu.cn (W.-C.L.); 1981318528@stu.hunau.edu.cn (X.-R.X.); liuyisong@hunau.edu.cn (Y.-S.L.); 2Orient Science & Technology College, Hunan Agriculture University, Changsha 410128, China; xlx5652397@163.com; 3Hunan Provincial the Key Laboratory of Protein Engineering in Animal Vaccine, College of Veterinary Medicine, Hunan Agricultural University, Changsha 410128, China

**Keywords:** genetic variation, phylogenetic analysis, ribosomal DNA, *Spirometra erinaceieuropaei*

## Abstract

Sparganosis is a neglected zoonotic parasitic disease that poses huge threats to humans worldwide. Snakes play an important role in sparganosis transmission because they are the most common second intermediate hosts for *Spirometra* parasites. However, the population genetics of *Spirometra* isolates from snakes is currently not well studied in China. The present study was performed to explore the molecular characteristics and phylogenetic analysis of *Spirometra* tapeworms from different species of snakes in Hunan Province. This study obtained 49 *Spirometra* isolates from 15 geographical areas in Hunan Province, Central China. Subsequently, the 18S and 28S ribosomal DNA (rDNA) fragments were amplified from the isolated parasites, and their sequences were analyzed to assess their genetic diversity. Phylogenetic analyses were performed using the maximum likelihood algorithm. The results showed that sequence variations among these isolates were 0–2.3% and 0–0.1% for 18S and 28S rDNA, respectively. The phylogenetic analysis showed that all *Spirometra* isolates from Hunan Province were clustered into the same branch with *Spirometra erinaceieuropaei* isolated from other areas (China, Vietnam, Australia). Moreover, the phylogenetic trees revealed that *Spirometra* is closely related to *Adenocephalus*, *Pyramicocephalus*, *Ligula*, *Dibothriocephalus*, *Schistocephalus*, and *Diphyllobothrium*. The *Spirometra* isolates of different hosts/regions in Hunan Province are not host segregated or geographically isolated, and support for the taxonomic status of *Spirometra* tapeworms in China has been added. These results provide reference values for future accurate identification and taxonomic status of *Spirometra* tapeworms in China.

## 1. Introduction

Human sparganosis is a worldwide disease caused by the larva (sparganum) of the genus *Spirometra* [1,2]. Humans can be infected through eating undercooked frog or snake meat and drinking polluted water [3,4]. Although sparganum has been reported to commonly reside in subcutaneous tissues and muscles, they can also migrate to the abdominal cavity, internal organs, eyes, and brain, which can form masses or space-occupying lesions in the body that cause local tissue damage and paralysis [5,6].

More than 10 species of the genus *Spirometra* have been reported, of which *Spirometra erinaceieuropaei* mainly infects humans. The first reported human case of sparganosis was discovered in 1882 by Patrick Manson from a man’s autopsy in Xiamen, and was named *Ligula mansoni* a year later [7]. *Sparganosis* has been mainly reported in China and can also be found in Europe (Poland, Italy, France, and the Czech Republic), Asia (Korea, Japan, Thailand, and Laos), South America (Ecuador, Paraguay, and Venezuela), and North America [5,8,9]. The reason for the high infection rate in China is mainly related to local customs. Superstitious people stick raw frog or snake flesh on skin wounds and even swallow tadpoles or snake bile in remote regions of China [4,10,11]. Another reason is the high infection rate of frogs and snakes in China. A survey showed that 14.3% (31/217) and 91.7% (344/375) of frogs and snakes, respectively, were infected in Hunan Province [11,12].

Although an important genus in zoonosis, the taxonomy of the *Spirometra* species has been controversial for a long time. It has also been suggested in some studies that the genus *Spirometra* belongs to the genus *Diphyllobothrium*, and should not form a separate genus [13,14]. Meanwhile, the valid species of *Spirometra* has also been unclear. This is still a mystery whether the pathogen of Chinese sparganosis is *S. erinaceieuropaei*, *Spirometra decipiens*, or both [11]. In the recent study of Yamasaki, it was found that two *Spirometra* species in Asia, neither of which is close to likely *S. erinaceieuropaei* originating from Poland, and lineage Type I is genetically diverse and widely distributed, however Type II is known so far only from Japan and Korea [15]. The primary and secondary ribosomal DNA (rDNA) structures remain stable during the long evolutionary process, which is one of the tools for studying phylogenetic evolution in parasites [16]. In the last few years of studies, ITS, 16S rDNA, 18S rDNA, and 28S rDNA have been used to establish the phylogenetic relationship of *Taenia* species [9,17,18,19,20,21]. The 18S and 28S rDNA contain both variable and conserved regions, which make them handy molecular markers to solve phylogenetic relationships at different levels [22]. This study analyzed the genetic diversity of the 18S and 28S rDNA sequences of *Spirometra* isolates from seven different hosts in 15 geographical regions in Hunan Province, and constructed the *Diphyllobothriidae* evolutionary tree. The main objectives of this study were as follows: (1) describe sample morphology; (2) perform a genetic diversity analysis of the collected isolates from different geographical locations and hosts in Hunan Province, China; and (3) investigate the taxonomic status of *Spirometra* isolates using 18S and 28S rDNA sequences from snakes in Hunan Province.

## 2. Materials and Methods

### 2.1. Sample Collection

This study collected 49 samples from the field site in 15 geographical locations of Hunan Province in Southern China between April and September 2018 (Table 1). Figure 1 provides a scheme of the geographical locations of the collected *Spirometra* tapeworms. *Spirometra* tapeworms were isolated from muscles and subcutaneous tissues of three snake species of the family *Colubridae*, i.e., *Ptyas dhumnades* (Cantor, 1842), *Elaphe carinata* (Günther, 1864), and *Elaphe taeniura* (Cope, 1861), as well as from the intestines of the family *Felidae*, i.e., *Panthera tigris* (Linnaeus, 1758), *Prionailurus bengalensis* (Kerr, 1792), *Felis silvestris* (Schreber, 1777), and feral domestic cats. The collected samples were then fixed in 70% ethanol and kept at −20 °C for the molecular analysis.

### 2.2. Morphological Observations

The live worms were washed by water three times, and then sprayed with heavy metal on the surface. The morphology was made using the SEM-6380LV scanning electron microscope (JEOL, Akishima, Japan). The scolex of the sparganum and the scolex, gravid proglottid, and egg of *Spirometra* tapeworms were directly glued to the sample table and sprayed with a gold coating, and photographs were taken using a JSM-6380LV scanning electron microscope.

### 2.3. DNA Extraction and Enzymatic Amplification

The total genomic DNA was extracted from individual samples using the Wizard^®^ SV Genomic DNA Purification System (Promega Corporation, Madison, WI, USA) following the manufacturer’s protocol. Two ribosome markers (18S and 28S rDNA) were amplified by polymerase chain reaction (PCR) using the primer combinations listed in Appendix A. PCR reactions were carried out in a 25 μL reaction mixture containing 8.5 μL distilled water, 12.5 μL Taq PCR Master Mix (Thermo Fisher Scientific, Waltham, MA, USA), 1 μL of each primer (25 pmol/L), and 2 μL DNA template in a thermal cycler (Biometra, Göttingen, Germany). For the 18S rDNA, the steps were 94 °C for 5 min (first denaturation) and five cycles of 96 °C for 1 min, 44 °C for 1 min, and 72 °C for 2 min, followed by 25 cycles with annealing temperature increased to 48 °C and then by 5 min at 72 °C (final extension). For the 28S rDNA, the steps were 94 °C for 5 min and 35 periods of 94 °C for 30 s, 50 °C for 30 s, and 72 °C for 1 min, followed by 72 °C for 5 min. A negative sample (no DNA) was used in each amplification run. Positive PCR products were purified and then sequenced in both directions by the Tsingke Company (Changsha, China).

### 2.4. Sequence Analysis

The obtained sequences in this study and the reference sequences were aligned using Clustal X 1.7 software [23]. The DAMBE v.5.2 program was used to measure the nucleotide substitution saturation [24]. In addition, the obtained sequences in this research were also compared with *S. erinaceieuropaei* isolates from Australia (*Canis familiaris*), Vietnam (*Xenochrophis flavipunctatus*), and China (*Amphiesma stolatum* and *Rana nigromaculata*) for 18S rDNA sequences, and Australia (*C. familiaris*), Vietnam (*X. flavipunctatus*), and China (*A. stolatum*) for 28S rDNA sequences, using the Megalign procedure in DNASTAR 5.0 software [25]. Moreover, DnaSP 5.0 was used to analyze the diversity indices (nucleotide diversity (π) and haplotype diversity (Hd)) of these three gene sequences obtained in the current research [26].

### 2.5. Phylogenetic Analysis

All of the sequences are aligned using Clustal W in MAGE7.0. The best nucleotide substitution models were selected using JModelTest0.1. Phylogeny was estimated using a maximum likelihood algorithm (ML) in MEGA7.0. The stability of the tree was calculated based on 1000 bootstrap replicates. Genetic relationships with other *Diphyllobothriidae* species as in-group and *Bothriocotyle solinosomum* as out-group were evaluated (Appendix B).

## 3. Results

### 3.1. Morphological Characteristics

In the scanning electron microscope study, the egg of *Spirometra* tapeworms was olive-shaped with slightly pointed ends and a slightly raised side, filled with many pores on the surface (Figure 2A–C). The scolex of the sparganum was flat, unsegmented, and with a wide front end, horizontal stripes, and apparent depression in the middle of the top end (Figure 2D–F). The adults were flat and segmented. The top of the adult scolex was sunken inward, and without other structure (Figure 2G). Moreover, many eggs existed in utero at the gravid proglottids (Figure 2H).

### 3.2. Genetic Characterisations of Spirometra Tapeworms

In this study, 49 and 49 PCR amplicons from 49 isolated samples were successfully amplified for 18S and 28S rDNA, respectively. No size differences were observed for any rDNA region among the amplicons tested (data not shown). The deletions and alignment lengths of the 18S and 28S rDNA were 2006–2010 and 1014 bp, respectively. The 28S rDNA target fragment amplified in this study is the front part of the entire 28S gene (highly protected area).

This study analyzed 49 18S sequences of *Spirometra* isolates. Intraspecific nucleotide variations within all isolates obtained in the present study were 0–2.3%. However, the 18S sequences obtained in the current study showed lower nucleotide variations of 0–1.6% compared with those of *S. erinaceieuropaei* from GenBank (China (KX528089 and HQ228991), Vietnam (KY552802), and Australia (KY552801). The pairwise comparison of the 28S rDNA sequences in the present paper showed 0–0.1% nucleotide variations. The sequence variation analysis for the 28S rDNA sequences showed higher nucleotide variations of 0–0.2% compared with those of *S. erinaceieuropaei* from GenBank (China (HQ228992), Vietnam (KY552835), and Australia (KY552836), and 0.60-0.90% compared with Diphyllobothriidea tapeworms (*Schistocephalus solidus*, *Diphyllobothrium scoticum*, *Diphyllobothrium sprakeri*, *Diphyllobothrium tetrapterum*, *Diphyllobothrium lanceolatum*, *Diphyllobothrium cordatum*, *Pyramicocephalus phocarum*, *Adenocephalus pacificus*, and *Ligula pavlovskii*).

The amplified 18S gene fragment sequence was 2006–2010 bp in length with 18 polymorphic sites. Moreover, insertions or deletions were found within the amplified fragments. Table 2 shows that the nucleotide diversity of the 18S sequences was 0.00062, which defined eight haplotypes with a haplotype diversity of 0.392. For 28S rDNA sequences (1014 bp), one polymorphic site was detected among 49 specimens examined in the present study, with no insertion or deletion. The diversity indices are shown in Table 2. The nucleotide diversity for the 28S rDNA sequences was 0.00021, defining two haplotypes with a haplotype diversity of 0.215.

### 3.3. Phylogenetic Relationship of S. erinaceieuropaei

A phylogenetic tree based on the 18S and 28S sequences was constructed using the ML method under the general time-reversible (GTR) model by MEGA7.0 (Figure 3). Data showed that all the isolated samples recorded in this study were grouped into one group, and clustered into the same branch with the *S. erinaceieuropaei* in Genbank from other countries (China, Vietnam, and Australia). In addition, a relatively complete phylogenetic *Diphyllobothriidae* tree was constructed based on the 18S and 28S sequences. In the current study, *Spirometra* spp. formed a separate group and were closely related to *Schistocephalus* spp. Moreover, the genus *Diphyllobothrium* occupied most of the phylogenetic tree, which was made up of *Adenocephalus* spp., *Pyramicocephalus* spp., *Ligula* spp., *Dibothriocephalus* spp., and *Schistocephalus* spp. However, the relationships among the species of *Diphyllobothrium* by 18S and 28S sequence were not established. *Duthiersia fimbriata* and *Duthiersia expansa* formed the *Duthiersia* spp. branch and then formed a sister group, the *Bothridium pithonis* branch.

## 4. Discussion

The species classification of *Spirometra* has been controversial. For many years, many researchers considered *S. erinaceieuropaei* as a global species [5,15]. As more and more mitochondrial gene sequences of *S. erinaceieuropaei* have been reported globally in recent years, studies have found that *S. erinaceieuropaei* in China and Southeast Asia and *S. erinaceieuropaei* in Europe do not belong to the same branch, which also means that the Chinese and Southeast Asia region may not be the previously thought *S. erinaceieuropaei* [7]. The present study aimed to analyze the genetic diversity of *Spirometra* tapeworms from snakes and to explore the taxonomic status of *Spirometra* isolates from Hunan Province on a molecular level. At the same time, this study provides the description of the morphology of *Spirometra* isolates from snakes in Hunan Province based on scanning electron microscopy, which will lay the foundation for future *Spirometra* tapeworm species classification in China.

The study used 18S and 2S rDNA genes to explore the intraspecific nucleotide variations of the *Spirometra* isolates in Hunan Province, China. The results show that the maximum variation values for the 18S and 28S rDNA sequences were 0–2.3% and 0–0.1%, respectively, among the *Spirometra* isolates from different hosts examined (*Zaocys dhumnades*, *Elaphe carinata*, *Elaphe taeniura*, *Panthera tigris*, *Prionailurus bengalensis*, *Felis silvestris*, and cat). The sequence variation analysis for the 18S gene showed 0–2.3% nucleotide divergence compared with those of *S. erinaceieuropaei* in China (*R. nigromaculata* KX528089 and *A. stolatum* HQ228991), Vietnam (*X. flavipunctatus* KY552802), and Australia (*C. familiaris* KY552801). This suggests that both host specificity and geographical effects are not the main factors contributing to the genetic variation of *S. erinaceieuropaei*, which can also be based on the results of the sequence variation analysis of 28S rDNA. This conclusion is in accordance with recently conducted research [9,21,27].

Haplotype and nucleotide diversities are two important indicators to measure the genetic variation of a gene. A base change can form a haploid type, and haploid type diversity can rapidly rise in a concise time. However, nucleotide base changes have little effect on nucleotide diversity. The rise of nucleotide diversity needs a long accumulation time. Thus, nucleotide diversity is more applicable for measuring the genetic diversity of a species [28]. For most organisms, a nucleotide diversity of >0.01 is considered a large variation [29]. In the current study, the nucleotide diversity of 18S and 28S rDNA genes of the *Spirometra* isolates was 0.00062 and 0.00028, respectively, which was lower than 0.01. The results showed that the genetic variation of *Spirometra* isolates from different hosts in Hunan Province was low.

In recent years, it has been shown by the molecular genetic evolution analysis that China and Poland are in different branches. Some scholars have proposed that *Spirometra* tapeworms should be restored to the title of *Spirometra mansoni* in China and Southeast Asia [7,15,30]. The phylogenetic tree based on 18S and 28S sequences showed that all the *Spirometra* isolates from different regions in Hunan Province formed a branch with *S. erinaceieuropaei* from Genbank from other countries (China, Vietnam, and Australia), except for the *S. erinaceieuropaei* reported in the United States. This result is consistent with Kuchta et al.’s proposal that China and Southeast Asia should be classified as *S. mansoni*, North America should be classified as S. decipiens, and Europe should be classified as *S. erinaceieuropai*. In the current study, phylogenetic trees revealed that *Spirometra* is closely related to *Adenocephalus*, *Pyramicocephalus*, *Ligula*, *Dibothriocephalus*, *Schistocephalus*, and *Diphyllobothrium* and forms a branch, which is similar to the study of Waeschenbach and Hernandez [18,21].

## 5. Conclusions

In our study, the genetic variability among different distinct developmental stages (larvae and adults) of *Spirometra* tapeworms isolated from 15 geographical areas in Hunan Province was analyzed for the 18S and 28S rDNA genes. The results revealed genetic variability in 18S and 28S rDNA. The phylogenetic tree based on 18S and 28S sequences revealed that the *Spirometra* isolates of different hosts/regions in Hunan Province are not host segregated or geographically isolated, and support for the taxonomic status of *Spirometra* tapeworms in China was thus added. These results provide reference values for future accurate identification and taxonomic status of *Spirometra* tapeworms in China.

## Figures and Tables

**Figure 1 vetsci-09-00062-f001:**
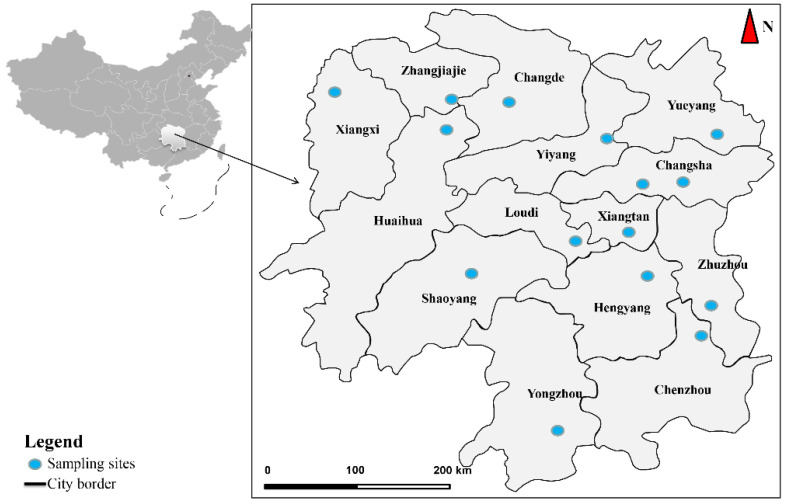
The sampling sites of *Spirometra* isolates in Hunan Province.

**Figure 2 vetsci-09-00062-f002:**
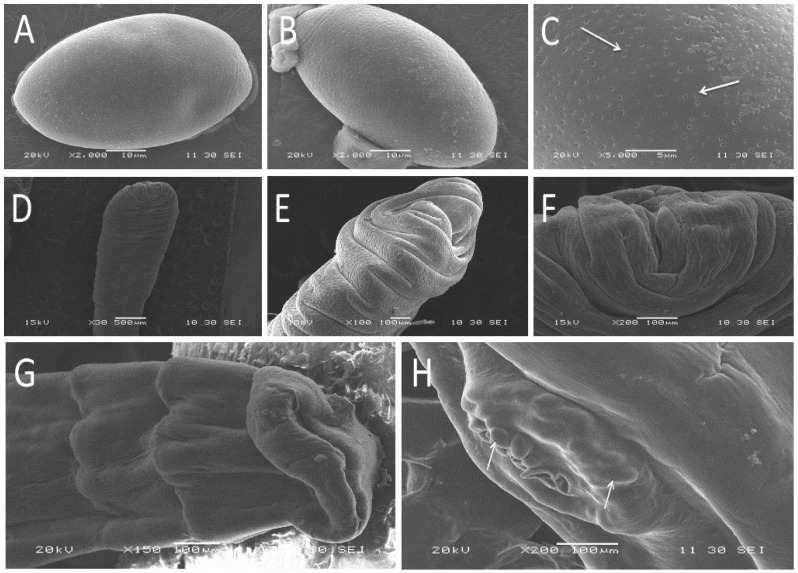
Scanning electron micrographs of *Spirometra* tapeworms collected from different hosts in Hunan Province, China. Egg (**A**,**B**). Detail of egg surface filled with pores (**C**). The scolex of larva, front view (**D**) and lateral view (**E**). Detail view of scolex (**F**). The scolex of adult (**G**). Detail view of egg in utero at the gravid proglottids (**H**).

**Figure 3 vetsci-09-00062-f003:**
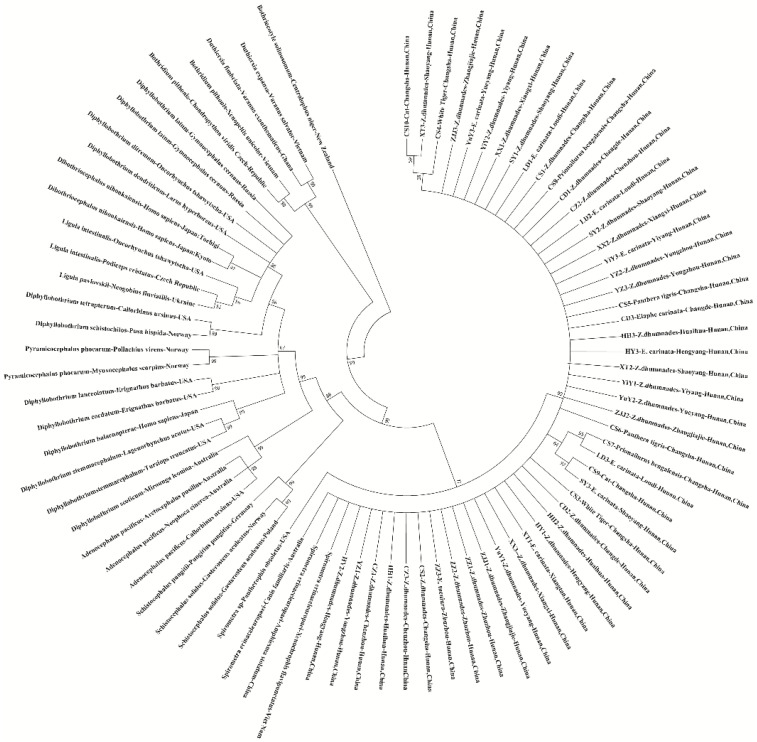
Maximum likelihood estimates of the phylogenetic relationships of *Spirometra* tapeworms based on 18S and 28S sequences computed in MEGA version 7.0.26 under the GTR model. The confidence levels in each node were assessed with the boot-strap method (1000 pseudo replicates) and bootstrap values >50.

**Table 1 vetsci-09-00062-t001:** Geographical origins (different locations in Hunan Province, China) of *Spirometra* tapeworms isolates used in this study, as well as their GenBank accession numbers for the 18S and 28S sequences.

Geographical Origins	Host	Location	Sample Codes
Yiyang City			
Lanxi Town, Heshan District	*Zaocys dhumnades*	112°46′ E, 28°59′ N	HuN-YiY1
	*Z. dhumnades*	112°46′ E, 28°59′ N	HuN-YiY2
	*Elaphe carinata*	112°46′ E, 28°59′ N	HuN-YiY3
Changde City			
Taizimiao Town, Hanshou County	*Z. dhumnades*	111°96′ E, 28°77′ N	HuN-CD1
	*Z. dhumnades*	111°96′ E, 28°77′ N	HuN-CD2
	*E. carinata*	111°96′ E, 28°77′ N	HuN-CD3
Yongzhou City			
Taiping Town, Ningyuan County	*Z. dhumnades*	112°13′ E, 25°67′ N	HuN-YZ1
	*Z. dhumnades*	112°13′ E, 25°67′ N	HuN-YZ2
	*Z. dhumnades*	112°13′ E, 25°67′ N	HuN-YZ3
Hengyang City			
Xuanzhou Town, Hengyang County	*Z. dhumnades*	112°85′ E, 27°24′ N	HuN-HY1
	*Z. dhumnades*	112°85′ E, 27°24′ N	HuN-HY2
	*E. carinata*	112°85′ E, 27°24′ N	HuN-HY3
Xiangtan City			
Jinshi Country, Xiangtan County	*Z. dhumnades*	112°75′ E, 27°59′ N	HuN-XT1
	*Z. dhumnades*	112°75′ E, 27°59′ N	HuN-XT2
	*E. carinata*	112°75′ E, 27°59′ N	HuN-XT3
Shaoyang City			
Shizhu Town, Dongkou County	*Z. dhumnades*	110°73′ E, 27°25′ N	HuN-SY1
	*Z. dhumnades*	110°73′ E, 27°25′ N	HuN-SY2
	*E. carinata*	110°73′ E, 27°25′ N	HuN-SY3
Zhuzhou City			
Jieshou Town, Chaling County	*Z. dhumnades*	113°43′ E, 26°61′N	HuN-ZZ1
	*Z. dhumnades*	113°43′ E, 26°61′N	HuN-ZZ2
	*Elaphe taeniura*	113°43′ E, 26°61′N	HuN-ZZ3
Changsha City			
Langli Town, Changsha County	*Z. dhumnades*	113°13′ E, 28°19′ N	HuN-CS1
	*Z. dhumnades*	113°13′ E, 28°19′ N	HuN-CS2
Changsha Ecological Zoo, Tianxin District	*White Tiger*	113°01′ E, 28°04′ N	HuN-CS3
	*W. Tiger*	113°01′ E, 28°04′ N	HuN-CS4
	*Panthera tigris*	113°01′ E, 28°04′ N	HuN-CS5
	*P. tigris*	113°01′ E, 28°04′ N	HuN-CS6
	*Prionailurus bengalensis*	113°01′ E, 28°04′ N	HuN-CS7
	*P.bengalensis*	113°01′ E, 28°04′ N	HuN-CS8
	*Cat*	113°01′ E, 28°04′ N	HuN-CS9
	*Cat*	113°01′ E, 28°04′ N	HuN-CS10
Loudi city			
Suoshi Town, Shuangfeng County	*E. carinata*	112°12′ E, 27°32′ N	HuN-LD1
	*E. carinata*	112°12′ E, 27°32′ N	HuN-LD2
	*E. carinata*	112°12′ E, 27°32′ N	HuN-LD3
Chenzhou City			
Longhai Town, Anren County	*Z. dhumnades*	113°29′ E, 26°48′ N	HuN-CZ1
	*Z. dhumnades*	113°29′ E, 26°48′ N	HuN-CZ2
	*Z. dhumnades*	113°29′ E, 26°48′ N	HuN-CZ3
Huaihua City			
Qijiaping Town, Yuanling County	*Z. dhumnades*	110°86′ E, 28°88′ N	HuN-HH1
	*Z. dhumnades*	110°86′ E, 28°88′ N	HuN-HH2
	*Z. dhumnades*	110°86′ E, 28°88′ N	HuN-HH3
Zhangjiajie City			
Dongxi Coutry, Cili County	*Z. dhumnades*	110°83′ E, 29°14′ N	HuN-ZZJ1
	*Z. dhumnades*	110°83′ E, 29°14′ N	HuN-ZZJ2
	*Z. dhumnades*	110°83′ E, 29°14′ N	HuN-ZZJ3
Yueyang City			
Tongshi Town, Pingjiang County	*Z. dhumnades*	113°72′ E, 28°75′ N	HuN-YuY1
	*Z. dhumnades*	113°72′ E, 28°75′ N	HuN-YuY2
	*E. taeniura*	113°72′ E, 28°75′ N	HuN-YuY3
Xiangxi City			
Xichehe Town, Longshan County	*Z. dhumnades*	109°54′ E, 29°09′ N	HuN-XX1
	*Z. dhumnades*	109°54′ E, 29°09′ N	HuN-XX2
	*Z. dhumnades*	109°54′ E, 29°09′ N	HuN-XX3

**Table 2 vetsci-09-00062-t002:** Diversity indices for *Spirometra* tapeworms using nucleotide data of the ribosomal 18S rRNA (2006–2010 bp) and 28S rRNA (1013 bp) gene sequences obtained in the present paper.

	N	S	H	π	Hd	K
18S	49	18	8	0.00062	0.392	1.244
28S	49	2	3	0.00028	0.275	0.281

N: number of isolates; S: number of polymorphic sites; H: number of haplotypes; π: nucleotide diversity; Hd: haplotype (gene) diversity; K: average number of nucleotide differences.

## Data Availability

Please refer to suggested Data Availability Statements at https://www.ncbi.nlm.nih.gov/nuccore/?term=18S+and+Spirometra+erinaceieuropaei and https://www.ncbi.nlm.nih.gov/nuccore/?term=28S+and+Spirometra+erinaceieuropaei.

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
