# Peer review of "Molecular Characterization and Phylogenetic Analysis of Spirometra Tapeworms from Snakes in Hunan Province"

_vetsci, 2022, doi:10.3390/vetsci9020062_

Round 1

Reviewer 1 Report

In this manuscript, Chen et al. presented systematic research about Spirometra tape-worms, including collection of live worms, SEM morphology study, genetic sequencing and phylogenetic analysis. Most importantly, the authors chose to use 18S and 28S rDNA sequences of to investigate the phylogenetic relationship and taxonomic status. 

The objective of this manuscript is very clear, and the methods for the research very sound. The data were unambiguously-presented, with novel findings to support the taxonomic status of Spirometra tapeworms in China.

Overall, this manuscript is highly recommended for publication. Especially considering the higher rate of infection in China and other Asian countries, Spirometra tapeworms should draw further attention from CDC and research institutes from these countries. (I personally come cross with news of such infections every year.)

Other suggestions:

Page 6, line 142, Figure 2(D) should be lateral view?

Reviewer 2 Report

This manucript aim to report the molecuar characterization of several Spirometra specimens retrieved from varios host from China. Although this work contributes with a large number of novel sequences from two molecular markers (18S and 28S) there are some issues to address before publication.

The introduction could be improve including the most resent taxonomic work on Spirometra such as Yamasaki et al. 2021.

Why the authors not amplified also COX1? There are several available Spirometra sequences from different regions of the world and could add better understanding of Spirometra taxonomy status.

If it is possible, it is strongly recommend to amplifiy cox 1 gene and include Spirometra  sequences from different parts of the world.

Reviewer 3 Report

  1. Page 1, Lines 2-3: Write in the title and throughout "tapeworms" as one word and not as "tape-worms".
  2. Page 1, Line 16: "posts huge threats"? Do you mean "poses huge threats"?
  3. Further I marked the pdf of the manuscript with numerous proposals for improvements. The most importan points follow:
  4. The Latin names of parasites and hosts should be in italics.
  5. the nomencalture of the snakes is not updated. I have included suggestions directly in the manuscript.
  6. Cestodes have no "mouthpart"!!! Say "forming a depression".
  7. Fig. 3 is very difficult to read! Use the entire width of the page for it or make another design.
  8. further linguistic and style suggestion are marked in the pdf.

Reviewer 4 Report

In the selected sentences, the names of species of parasites should be used in italics:

Line 46, 48, 60, 82, 83

Table 1. "Host"

Line 117, 118, 119, 120

Line: 154, 158, 160-163

Line: 181, 184, 185, 186-190

Line: 198-202, 212-217, 219, 234, 237-243

Round 2

Reviewer 2 Report

The authors addressed the comments and improved the introduction.

I believe the manuscript is ready for publication.